# Multiscale Permutation Lempel–Ziv Complexity Measure for Biomedical Signal Analysis: Interpretation and Application to Focal EEG Signals

**DOI:** 10.3390/e23070832

**Published:** 2021-06-29

**Authors:** Marta Borowska

**Affiliations:** Faculty of Mechanical Engineering, Bialystok University of Technology, 45C Wiejska St., 15-351 Białystok, Poland; m.borowska@pb.edu.pl; Tel.: +48-694-424-442

**Keywords:** multiscale approach, permutation Lempel–Ziv complexity measure, focal EEG signals

## Abstract

This paper analyses the complexity of electroencephalogram (EEG) signals in different temporal scales for the analysis and classification of focal and non-focal EEG signals. Futures from an original multiscale permutation Lempel–Ziv complexity measure (MPLZC) were obtained. MPLZC measure combines a multiscale structure, ordinal analysis, and permutation Lempel–Ziv complexity for quantifying the dynamic changes of an electroencephalogram (EEG). We also show the dependency of MPLZC on several straight-forward signal processing concepts, which appear in biomedical EEG activity via a set of synthetic signals. The main material of the study consists of EEG signals, which were obtained from the Bern-Barcelona EEG database. The signals were divided into two groups: focal EEG signals (*n* = 100) and non-focal EEG signals (*n* = 100); statistical analysis was performed by means of non-parametric Mann–Whitney test. The mean value of MPLZC results in the non-focal group are significantly higher than those in the focal group for scales above 1 (*p* < 0.05). The result indicates that the non-focal EEG signals are more complex. MPLZC feature sets are used for the least squares support vector machine (LS-SVM) classifier to classify into the focal and non-focal EEG signals. Our experimental results confirmed the usefulness of the MPLZC method for distinguishing focal and non-focal EEG signals with a classification accuracy of 86%.

## 1. Introduction

Electroencephalography (EEG) is a non-invasive procedure to measure the bioelectric activity of the brain. The analysis of EEG signals contributes to a better understanding of brain functions and malfunctions. Electroencephalographic signals from epilepsy patients are of interest to researchers in particular.

Epilepsy is one of the most common chronic neurological diseases, characterized by recurrent seizures that can be single or complex in nature and are manifested by disturbances in consciousness, behavior, perception, movement, or sensation. According to WHO data [1], over 50 million people suffer from epilepsy, and every year, an average of 2.4 million patients are diagnosed with epilepsy. If an appropriate diagnosis and treatment are made, 70% of the patients with disease can live without seizures. The clinical picture of the seizure is a result of a discharge of improperly synchronized neurons, and it is influenced by the size of the abnormally stimulated area and its location. In 2017, the current seizure classification was published [2], which includes the division into focal seizures, generalized seizures, and seizures with unknown onset. In the focal-onset seizures category, there are seizures with and without the state of consciousness, with the coexistence of motor symptoms and the development of focal seizures to bilateral tonic-clonic seizures. Among generalized seizures, there are motor seizures and non-motor seizures (absences). In the case of seizures with unknown onset, there are seizures with or without the coexistence of motor symptoms as well as unclassified seizures. The new classification noted that some tonic and flexion seizures may also have a focal onset. International League Against Epilepsy (ILAE) also presented a new classification of epilepsy and epilepsy syndromes that takes into account the following diagnostic criteria: seizure type, epilepsy types, and epilepsy syndromes [2].

The diagnosis of epilepsy patients sometimes requires EEG recording directly from the brain surface or from deeper brain structures [3]. Intracranial records can be used to locate the areas of the brain where seizures begin and to evaluate the benefits of neurosurgical resection of these parts of the brain. It turns out that the recordings show variable dynamics not only during acute epileptic seizures but also in the non-seizure period. There is a need to develop new methods for analyzing signals from epilepsy patients in the seizure-free period to identify them at an early stage of diagnosis to implement the appropriate treatment method as soon as possible.

Nonlinear analysis of EEG signals includes many measures that allow for extraction of useful information from dynamical systems. There are many methods of detecting dynamic changes in physiological systems; some complexity indexes in particular, such as Lempel–Ziv complexity [4], permutation Lempel–Ziv complexity [5], approximate entropy [6], sample entropy [7], fuzzy entropy [8], permutation entropy [9], multi-scale entropy [10,11,12,13], recurrence quantification analysis [14], detrended fluctuation analysis [15], and fractal dimension [16] are used as effective features of EEG signals [17,18,19,20,21,22,23,24].

Related researches have indicated that LZC [4] is powerful in analysing biomedical signals, especially in EEG analysis [25,26,27,28,29]. LZC based on a coarse-graining process is a nonlinear measure of signal complexity and irregularity for short and nonstationary time series. Higher value of LZC implies a more complex structure of the signal. The original Lempel–Ziv complexity algorithm consists of transformation of the signal into a binary sequence by comparing it with the threshold (e.g., mean or median) and calculating the unique subsequence in a sequence. However, the LZC measure is artifact sensitive and cannot distinguish between deterministic chaos and noise [5,30,31]. The binary coarse-graining process is associated with loss of signal dynamics and important system information. Ordinal patterns [9] have been used to quantify dynamical information of signals as an improvement of binary coarse-graining process. Bai et al. [5] developed permutation Lempel–Ziv complexity measures (PLZC) to quantify dynamic changes in EEG signals. However, the existing PLZC methods have one common problem when analyzing EEG signals: EEG information is embedded in different scale domains. PLZC algorithm is a single-scale analysis, and therefore, it fails to account for multiple electrical activities that are inherent in the brain. Using the multiscale approach to measure the complexity of EEG recordings over multiple time scales of signals instead of using a single scale [10] is a solution to this problem. Therefore, we proposed the multiscale permutation Lempel–Ziv complexity measure (MPLZC) that combines permutation Lempel–Ziv complexity methods and a multiscale approach. The concept of multiscale Lempel–Ziv measure has already been introduced in the analysis of short, non-stationary, and noisy EEG signals [32]. However, this method uses multiple thresholds for binarization, which is obtained by comparing each element of EEG signals with its smoothed versions in the window.

The purpose of the research presented in this article is to evaluate MPLZC measure in identification of focal signals.

The structure of the paper is organized as follows: Section 2 presents the description of simulated and empirical data, methods of Lempel–Ziv complexity, permutation Lempel–Ziv complexity, multiscale permutation Lempel–Ziv complexity, and the classification. The results are summarized in Section 3. Finally, Section 4 presents the discussion of the obtained results with corresponding analysis.

## 2. Materials and Methods

The Python language, an interpreted high-level performance programming language, was used for the analysis of the synthetic signals and EEG signals.

### 2.1. Simulated Data

In this subsection, the simulated signals used to evaluate the MPLZC measure in terms of classical signal-processing concept (frequency, amplitude, noisy power, signal bandwidth) are described. Some of them have been used to evaluate the Lempel–Ziv complexity measure [26], auto-mutual information function [33], and composite multiscale permutation entropy [34]. These generated signals have a sampling frequency ($f_{s}$) of 100 Hz and a length of 100 s. Additionally, white Gaussian noise (WGN) and 1/f noise were used to verify the performance of MPLZC measures. The simulated signals used are listed as follows:Gaussian noise

White Gaussian noise is a random signal having equal intensity at different frequencies, giving it a constant power spectral density S(f):(1)$$S\left(f\right)=\frac{c_{w}}{{\left|f\right|}^{0}}$$
where $c_{w}$ is constant. It is a discrete signal whose samples are regarded as a sequence of serially uncorrelated random variables with zero mean and finite variance (Figure 1a).

Pink noise, or 1/f noise, is a signal or a process with such a frequency spectrum that the power spectral density S(f) (power-per-frequency interval) is inversely proportional to the frequency of the signal:(2)$$S\left(f\right)=\frac{c_{f}}{{\left|f\right|}^{\alpha }}$$
where $c_{f}$ is constant, and 0<α<2. The complexity of pink noise is greater than that of white noise, with pink noise being more regular and predictable than white noise [35] (Figure 1b).

Each type of noise was generated 50 times.

Sinusoidal signals with variable amplitude and frequency

Sinusoidal signals with variable amplitude and frequency were generated in order to explain how the MPLZC change when the amplitude and frequency of sinusoidal signals are changed. The first signal consists of constant amplitude of the chirp signal, with a logarithmically variable frequency in the range of 0.1 Hz to 20 Hz in 100 s (Figure 2). In the second signal, the frequency changes logarithmically from 0.25 Hz to 5 Hz in 100 s with chirp amplitude modulation of the signal (Figure 2). The signal analysis was performed using a window.

Amplitude modulated quasi-periodic signal with the addition of WGN of diverse power

Amplitude modulated quasi-periodic signals with the addition of WGN of diverse power were used to verify how the MPLZC change with the level of noise. Amplitude modulated quasi-periodic signals were generated as a sum of two sinusoidal waveforms with the frequencies of 0.5 Hz and 1 Hz. The first 20 s of the resulting signal do not contain noise. After that time, white noise was added to the signal with increasing power every 10 s (Figure 2).

Bandwidth of coloured Gaussian noise signal

The bandwidth of coloured Gaussian noise signal was generated in order to determine the relationship between MPLZC and the noise bandwidth. The signal consists of five segments of coloured noise in different bands. The color noise frequency spectra are centered at fs/4, and their bandwidth increases from fs/15 to fs/3 in five equal steps (Figure 2).

Signal with spectral colour noise content

The signals are generated to investigate the relationship of the MPLZC and the spectral content of the colour noise using an autoregressive process of order 1 for the time table t with the parameter going from +0.9 to −0.9 linearly (Figure 2).

Periodic deterministic process

The MIX process is defined as:(3)MIX=(1−z)x+zy
where z is a random variable equal to 1 for probability *p* and equal to 0 for probability 1−*p*, *x* is a periodic time series made of $x_{k}=\sqrt{2}\sin\left(\frac{2\pi k}{12}\right)$, and y is a random variable with a distribution in the range $\langle -\sqrt{3},\sqrt{3}\rangle$. A synthetic time series is based on the MIX process for parameters between 0.01 and 0.99 (Figure 2).

### 2.2. Data Collection

In the present work, we used Bern-Barcelona EEG database (https://www.upf.edu/web/ntsa/downloads, accessed on 1 March 2021) [3] to verify the usefulness of MPLZC measure for long-term intracranial electroencephalographic recordings (EEG) from patients suffering from pharmaco-resistant, focal-onset epilepsy. The clinical purpose of those recordings was to determine the brain areas to be surgically removed in each individual patient in order to achieve seizure control. The EEG data contains 3750 pairs of signals recorded from the areas of the brain where the first ictal EEG signal changes were detected (focal signals) and 3750 pairs of signals recorded from the areas of the brain that were not involved at seizure onset (non-focal signals) obtained from five temporal-lobe epilepsy patients. Multichannel EEG signals were recorded with intracranial strip and depth electrodes manufactured by AD-TECH (Racine, WI, USA). All EEG signals were digitally band-pass filtered between 0.5 and 150 Hz using a fourth-order Butterworth filter. The sampling frequency was 512 Hz. The length of each signal was 10,240 samples. The recordings of seizure activity and recordings made three hours after the last seizure were excluded. Each individual signal pair was selected randomly from the pool of all signals measured at focal EEG channels. This random sample was drawn without replacement and using a uniform random number generator. In the same way, non-focal signals measured at non-focal EEG channels were randomly selected. Focal and non-focal signals came from independent areas of the brain. The first 50 pairs of records were used for that analysis and were made to form 100 signals in the focal and non-focal group.

### 2.3. Lempel–Ziv Complexity

The Lempel–Ziv complexity measure can characterize the degree of order or disorder and development of spatiotemporal patterns [4]. The signal $x\left(n\right)=\left\{x_{1},x_{2},\ldots ,x_{n}\right\}$ must be converted by a coarse-graining process into a finite sequence {s(n)} whose elements contain zeros and ones. The coarse-graining process is very important because it determines how much information can be preserved from the original signal. A commonly used coarse-graining method is to select threshold $T_{d}$ as the mean value of the window time series and transform the original signal into a 0−1 sequence by comparing the signal with $T_{d}$. Another way is to adopt a method using the Hilbert transform approach to generate a 0−1 sequence [16,17]. The Lempel–Ziv algorithm counts all distinct patterns in a sequence {s(n)}. The complexity counter c(n) increments by one when a new pattern is found. The LZC measure has a range from 0 to 1 after normalization, where 0 means order, and 1 means random pattern. The calculation of c(n) can be represented as follows:

Step 1: Transform the signal into a finite sequence {s(n)} that contains zeros and ones.

Step 2: Let A and B denote a subsequence of the sequence {s(n)}.

Step 3: Connect A and B into AB. Sequence ABc is derived from AB and c (c means that the last digit has to be deleted). Suppose that A=s(1),s(2),…s(k), B=s(k+1); then ABc=s(1),s(2),…s(k). Let f(ABc) express the vocabulary of all different subsequences of ABc.

Step 4: The rule if B belongs to f(ABc) or not. If B∈f(ABc), then B is not a new sequence, and A does not change. Update B as s(k+1),s(k+2). If B=s(k+1),s(k+2),…s(k+i) and is not a subsequence of ABc=s(1),s(2),…,s(k),s(k+1),…s(k+i−1), then B is a new sequence, and A is changed. Update B as B=s(k+i+1) and A as A=s(1),s(2),…s(k),s(k+1),…s(k+i). The complexity counter c(n) is incremented by one. Go to step 3.

Step 5: c(n) means the number of distinct patterns of sequence {s(n)}. Normalized Lempel–Ziv complexity can be defined as:(4)$$C\left(n\right)=\left(c\left(n\right)\;\log_{2}n\right)⁄n$$

### 2.4. Permutation Lempel–Ziv Complexity

The permutation Lempel–Ziv complexity measure combines the permutation scheme and Lempel–Ziv complexity [5]. In the first step of PLZC, a finite sequence of symbols {s(n)} was generated that included a total of m! types of symbols, where m is the number of data points in each motif. The permutation Lempel–Ziv complexity measure depends on an order m and time delay τ. Given a scalar time series $x\left(n\right)=\left\{x_{1},x_{2},\ldots ,x_{n}\;\right\}$, a vector $y_{m}\left(N\right)=\left[y_{N},y_{N+\tau },\ldots ,y_{N+\tau \left(m-1\right)}\right]$ composed of the m-th subsequent values is constructed, where m is the number of samples belonging to the subsequence, and τ represents the distance between the samples spanned by each section of the motif. Then, the permutation $s=\left(s_{0},s_{1},\ldots ,s_{\tau \left(m-1\right)}\right)$ is defined as an ordinal pattern associated with the vector $y_{m}\left(N\right)$, which is arranged in increasing order: $y_{N+s_{0}}\leq y_{N+s_{1}}\leq \cdots \leq y_{N+s_{\tau \left(m-1\right)}}$. These symbols are used in the procedure of calculation of the Lempel–Ziv complexity measure, which is normalized as follows:(5)$$PLZC=\left(c\left(n\right){log}_{m!}n\right)⁄n$$

### 2.5. Multiscale Permutation Lempel–Ziv Complexity

MPLZC combine two topics. First, a coarse-graining procedure is applied to the original time series $x\left(n\right)=\left\{x_{1},x_{2},\ldots ,x_{n}\right\}$. The coarse-grained time series are constructed in non-overlapping windows of increasing length s (called scale factors) for which the number of data points is averaged. Each element of the coarse-grained time series is calculated as follows:(6)yjs=1s∑i=(j−1)s+1jsxi 1≤j≤⌊Ns⌋
where ⌊a⌋ denotes the largest integer not greater than a. The length of each coarse-grained time series is s times shorter than x(n) (s=1→the original time series). Next, PLZC is calculated for each scaled series and plotted as a function of the scale factor s. The coarse-grained process uses a procedure similar to sub-sampling.

### 2.6. Classfication

The support vector machine (SVM) algorithm can be used for both classification and regression [19]. The SVM method transforms the original space in which the problem of classification has been defined to *n*-dimensional space (*n*-the number of features). The transformation is made in such a way that after it is made in the new space, objects are separable by means of a hyperplane (this separation is usually impossible in the original space). The hyperplane can be found using the least squares support vector machine (LS-SVM). The main element of the transformation is the selection of the kernel function responsible for mapping the points to the new space. The SVM algorithm works very well in practical applications, such as biomedical data analysis.

LS-SVM classification was applied to confirm the usefulness of the obtained twenty MPLZC features. Scikit-Learn tool [36] implemented in Python was used to measure the accuracy performance of a LS-SVM classifier, and traditional C-support vector classification (C-SVC) was used as the support vector classifier. The final classification accuracy is the mean result of the ten-fold cross-validation procedure. The radial basis function (RBF kernel) was used as a kernel function. The performance of the LS-SVM classifier was evaluated using sensitivity (*SEN*), specificity (*SPF*), and accuracy (*ACC*). The mathematical expressions of the classification parameters are presented as follows [32,37,38]:(7)SEN=(TP/(TP+FN))×100%
(8)SPF=(TN/(TN+FP))×100%
(9)ACC=((TP+TN)/(TP+TN+FP+FN))×100%

## 3. Results

### 3.1. Parameter Selection

The unity delay (τ=1) is used to calculate the PLZC of the coarse-grained time series for all scales. There are several methods for determining the optimal time delay for a single channel. Future developments should test time delay selection for different scales, but these are not yet available.

The order m is the number of data points contained in the motif. In the permutation process, m can be given a value of 3, 4, 5, 6, or 7 [5,9]. When m<3, there would be too few possible patterns, and for m>7, the PLZC algorithm generates m! possible motifs, causing a large computational cost. The condition m!≤N−(m−1) must hold so that every possible order occurs in the signal. Based on our experience, we recommend m=4 for N ≥ 1000 or m=5 for N ≥ 2000, as the value is usually less than m!. Additionally, the condition of multi-scale analysis $\left(m+1\right)!\leq \lfloor \frac{N}{s}\rfloor$ should be kept.

### 3.2. Synthetic Signals

Firstly, MPLZC analysis was applied to simulated signals: white Gaussian noise, 1/f noise signal (Figure 3), chirp signal with constant amplitude, signal with logarithmic chirp and AM modulation, amplitude-modulated quasi-periodic signal with additive WGN of diverse power, and coloured Gaussian noise that is appended one after the other, with different bandwidth, autoregressive of order 1; MIX signal (1−z)x+zy includes a periodic and a stochastic process [34] (Figure 4). WGN and 1/f noise, as two commonly used signals in multiscale analysis, were used. As it can be seen in Figure 3 for WGN, MPLZC values increased monotonically with scale factor s. That tendency was maintained at different lengths of the analyzed signal except for the signal with a length of 256 samples (=2.5 s). It could result from an incorrectly selected parameter m. In our study, we choose a low dimension m = 4 and N = 1000 when calculating MPLZC. MPLZC values for WGN signal and 1 / f noise decrease with increasing signal length on all scales, and the standard deviation of the average of 50 signals results decreases. MPLZC values for 1/f noise increase and become constant for higher scales.

In Figure 4, the results of MPLZC measure on synthetic signals are shown. A sliding window moving for 10 s with 80% overlap with the objective of testing was used to understand the relationship between MPLZC and frequency, amplitude, noise power, or signal bandwidth.

The result of MPLZC show that values increase for chirp signals with constant amplitude as the frequency of signal increases, and amplitude modulation of this signal has no significant effect on the results compared to constant amplitude signal, as it can be observed in Figure 4a,b, respectively.

Figure 4c shows the relationship between MPLZC and changes in additive noise power in quasi-periodic signals. MPLZC values increase as the power of the noise increases. MPLZC is sensitive to changes in noise power. MPLZC values are lower for higher scales; a multiscale process causes signal filtering.

Figure 4d shows MPLZC results in reference to increasing the noise bandwidth. The signal consists of four segments of coloured noise with increasing spectral bandwidth. MPLZC is sensitive to signal bandwidth changes, especially when scale factor s<5. MPLZC values increase along the signal. When scale factor s≥5, MPLZC takes similar high values.

Figure 4e shows the relationship between MPLZC and an autoregressive process. MPLZC for small scale increases along the signal and, for higher scale, takes higher values.

Figure 4f shows the results of MPLZC for MIX process, which evolves from randomness to periodic oscillations. MPLZC values decrease along the signals for all scales s, especially when scale factor s=1, s=2, and s=3.

### 3.3. Neurological Focal—Non-Focal Dataset

In this study, the MPLZC method was applied to analyze the EEG signals. Figure 5 shows an example of the bivariate signals from the focal and non-focal group.

Permutation Lempel–Ziv complexity was evaluated for 20 scale factors with the dimension m = 4. Mann–Whitney rank test (the non-parametric equivalent of t-student statistics for the independent samples) was used to find significant differences between the groups, and results were considered statistically significant for *p*-values < 0.05. The result of the MPLZC analysis on the averaged EEG signals is shown in Figure 6. For scale 1, which is regarded as a single-scaled base method, and for other scales, the mean PLZC values of groups are presented in Table 1. The permutation Lempel–Ziv complexity values in the focal group are significantly lower than those in non-focal group for scales above 1 (*p* < 0.05), which indicates that non-focal signals are more complex/irregular. The permutation Lempel–Ziv measures increase for all scales. Standard deviation values for both groups are large, but the non-focal group shows higher values of them, as can be seen in the Figure 6. The strongest separation between the focal group and non-focal group occurs for higher scales, which can improve proper identification of the focal area.

Comparing the signals shown in Figure 7, it can be seen that the signal dynamics of focal and non-focal areas are different. Focal signals include a certain repetition of patterns, while the non-focal signals are more irregular. In this work, differences in the complexity of both signals are shown by the MPLZC method. The lower values of MPLZC obtained for focal group indicate that the dynamics of focal signals are more regular than for non-focal signals. This seems typical because focal signals are more periodic and less random.

The result of the SVM classification is presented in Table 2. The MPLZC features achieve accuracy of classification up to 86% (confidence intervals ±5%).

## 4. Discussion

The evolution of computational techniques has enabled the development of non-linear methods of analysis. These methods are derived from chaos theory and are often used in the study of complex biological systems. Chaos means some kind of disorder/irregularity in which there are both deterministic and stochastic elements. Lempel–Ziv complexity measure borrowed its useful approach for estimating randomness of finite symbolic sequence from chaos theory. Higher LZ values indicate the presence of new sequence patterns and thus a more complex dynamic behavior. The Lempel–Ziv measure shows the rate at which new patterns appear with respect to the underlying system regardless of whether the system is deterministic or stochastic.

The study of dynamic systems in medicine characterized by high sensitivity to initial conditions suggests that the human body shows the highest degree of complexity. Living organisms are affected by changing internal and external factors. In order to maintain an actively changing body balance, it needs regulatory mechanisms (homeodynamics) based, inter alia, on the feedback loop. These dynamic systems, in a healthy person, contain oscillations of physiological parameters in acceptable ranges. The normal EEG signal consists of disorganized fluctuations, but during epileptic seizure, the EEG signals are more rhythmic. The change of such states is reflected by complexity. The LZ algorithm describes the EEG signal in the time domain, but there are studies showing the sensitivity of the method to changing the power spectrum and amplitude distribution in the time domain. Therefore, it seems justified to use LZ in the analysis of EEG signals. The Lempel–Ziv measure is based on the procedure of converting a signal into a binary signal using differently determined threshold values (e.g., mean, median). Unfortunately, during such conversion, information from the original signal is lost. The study [32] showed that classic LZC has low sensitivity in the assessment of changes of small amplitude. Bai et al. [5] proposed a new PLZC algorithm that combines the permutation with the Lempel–Ziv complexity. They showed the ability to differentiate clinical conditions in EEG recordings. The PLZC reflects the relationship of signal points (described as motifs) to one another, and the variability of these motifs indicates a change in the signal itself. The changes will be visible in both high and low frequency. However, PLZC is limited to assessing the values for only one temporal scale. This technique fails to account for multiple time scales inherent in biomedical signals. To deal with this limitation, a multiscale approach [10] was used to construct the new procedure of multiscale permutation Lempel–Ziv complexity measure.

MPLZC values for noisy signals are higher than for periodic signals. As the length of the analyzed signal increases, the variability of the results (the standard deviation) decreases. The results were more stable. For both constant and amplitude-modulated chirp signal, the MPLZC values were increased along the windowing signal until the highest values were reached. The transitions between the windows were quite smooth. The analysis showed that there were no significant differences in slow change in amplitude. In the case of a quasi-periodic synthetic signal, the MPLZC values generally increased in successive time windows. For a scale factor of 1, the amount of noise power was not recognizable from the PLZC values because they were saturated. However, given the higher scale factors, the importance of the multiscale concept can be seen. Different noise power levels saturated a different number of temporal scales. For coloured noise, MPLZC was sensitive to signal bandwidth changes with increasing bandwidth. The analysis in AR(1) process showed that a part of the signal with wider spectra has higher values. Mix process showed that values of MPLZC decrease from randomness to periodic oscillations.

By using the MPLZC measure, the EEG signal analysis is focused on the possibility of differentiation of focal from non-focal signals. Lower PLZC values of focal signals suggest that the arrangement of brain EEG signals increases during focal epilepsy. This is consistent with the general hypothesis that reduction in the permutation Lempel–Ziv complexity of biological signals is associated with disease [10]. Moreover, the t-student test has shown that the MPLZC values of focal signals are statistically higher than that of non-focal signals for scales above 1 (s > 1). In this study, nonlinear MPLZC features were extracted from EEG signals. Using 20 MPLZC features, a maximum accuracy of up to 86% was obtained. The published algorithms based on the entropy features for detection of focal signals are summarized in Table 2. Most studies employed various decomposition techniques prior to the extraction of entropy features. Some of them used entropy combination and other nonlinear features to characterize focal and non-focal signals. During the analysis, the measure of LZC was calculated, but it did not show any significant differences in mean values between the groups in any study. The use of a multiscale approach allowed for achieving a satisfactory result.

Related studies have indicated that complexity indexes are powerful in analyzing focal versus non-focal signals within the Bern-Barcelona EEG database (Table 3). The presented results seem to suggest that the main classification method is the support vector machine (SVM). Sharma et al. [19] used various entropy measures, such as Shannon entropy, Renyi’s entropy, approximate entropy, sample entropy, and phase entropy, from the decomposed EEG signals using empirical mode decomposition [19], discrete wavelet transform [18], and the tunable-Q wavelet transform [20]. Their group achieved accuracy of classification at the level of 84%, 87%, and 95%, respectively. Das et al. [39] proposed combined EMD and DWT methods to decompose EEG signals and used entropy measures from the decomposed coefficients, achieving the classification accuracy of 89%. Bhattacharyya et al. [40] adopted multivariate fuzzy entropy of the sub-band signals obtained using TQWT. The proposed method achieves the highest classification accuracy of 84.67%. Acharya et al. [24] presented a literature review on the classification of focal signals and achieved a classification of 87.93% using 23 nonlinear features. Gupta et al. [41] used EMD along with Sharma–Mittal entropy feature computed on Euclidean distance values from K-nearest neighbors (KNN). This new methodology allows for achieving the classification accuracy of 83.18%. The results obtained show a similar accuracy of the classification.

The main highlights of this study are as follows:This is the first study to introduce a new methodology of complex systems analysis;The features obtained from MPLZC analysis allow for distinguishing two classes of EEG signals;We used only one measure, so it can be useful in building a real system supporting identification of a epileptogenic activity in an area of the brain.

In the feature, the proposed solution should be applied with reconstructed source-space data from surface EEG, EMG (electromyography), or EHG (electrohysterography).

## Figures and Tables

**Figure 1 entropy-23-00832-f001:**
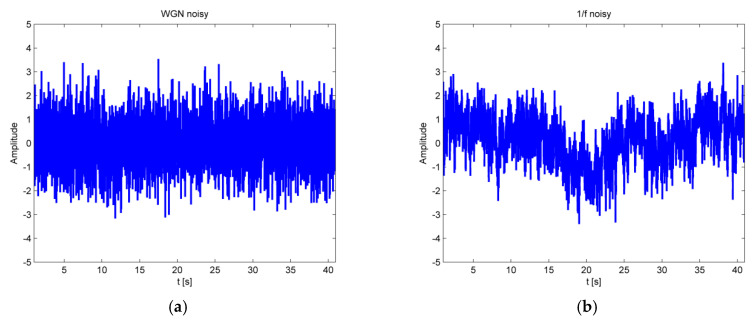
Example of simulated signals: (**a**) white Gaussian noise; (**b**) 1/f noise.

**Figure 2 entropy-23-00832-f002:**
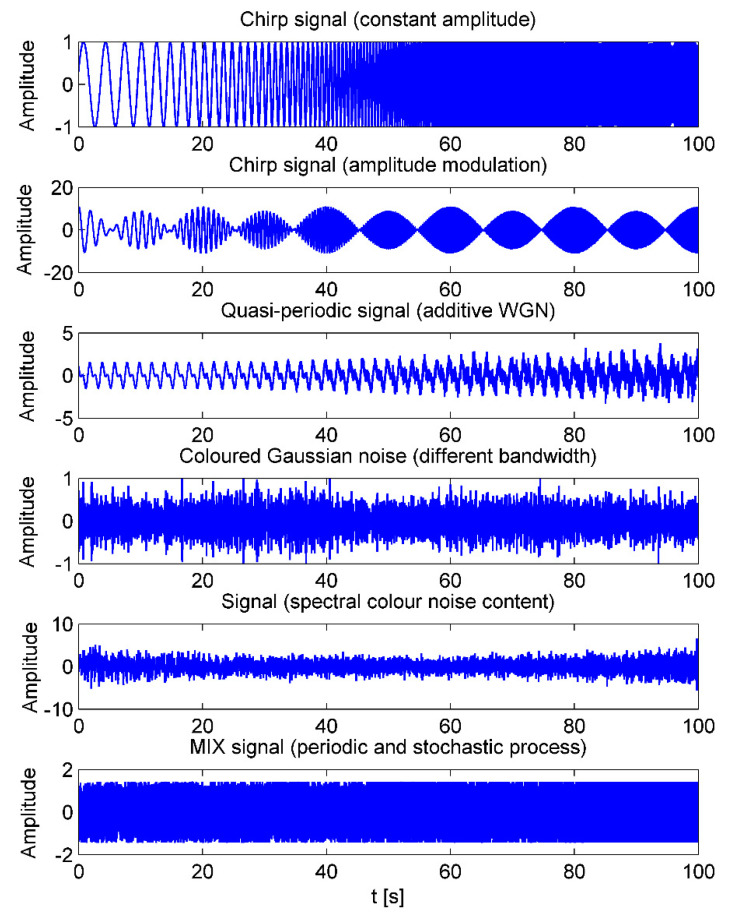
Examples of simulated signals: chirp signal with constant amplitude, signal with chirp amplitude modulation of the signal, quasi-periodic signal with additive WGN of diverse power, and coloured Gaussian noise with different bandwidth, spectral content of the colour noise using an autoregressive process of order 1; MIX signal (1−z) x+zy includes a periodic and a stochastic process [34].

**Figure 3 entropy-23-00832-f003:**
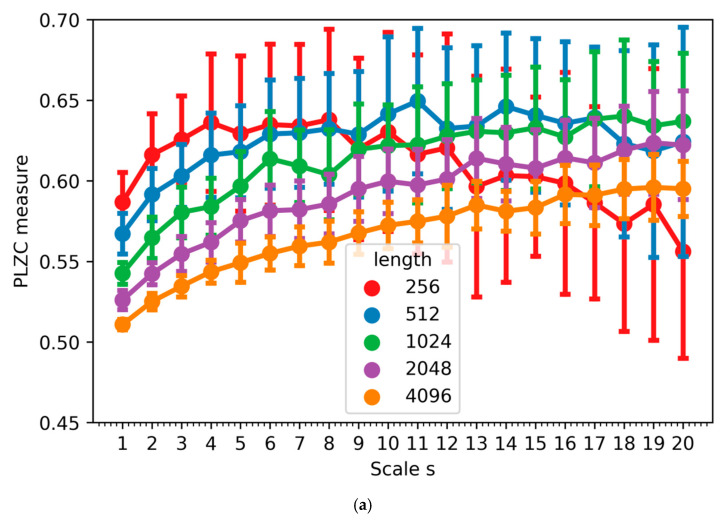
The result of MPLZC analysis for: (**a**) WGN noise; (**b**) 1/f noise.

**Figure 4 entropy-23-00832-f004:**
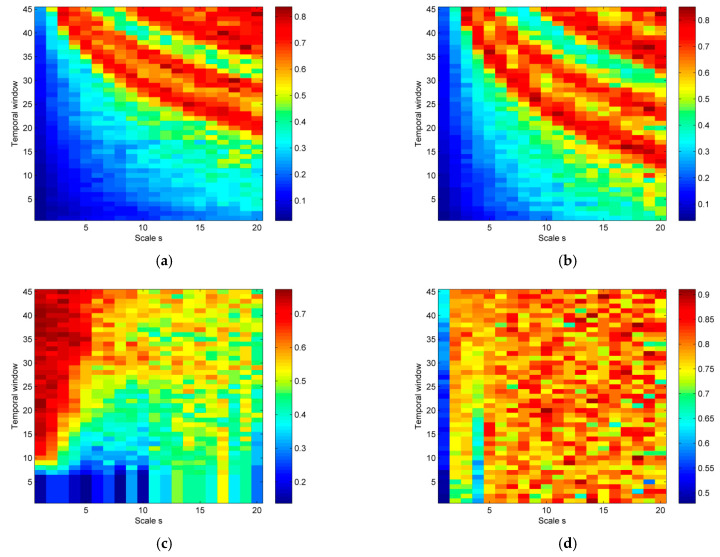
The results of the MPLZC measure on synthetic signals. The relationships between: (**a**) MPLZC and chirp signal with constant amplitude, (**b**) MPLZC and amplitude-modulated chirp signal, (**c**) MPLZC and quasi-periodic signal with increasing additive noise power, (**d**) MPLZC and a signal including five segments of coloured noise with increasing bandwidth, (**e**) MPLZC and AR(1) process with variable parameter, and (**f**) MPLZC and a MIX process which evolves from randomness to periodic oscillations.

**Figure 5 entropy-23-00832-f005:**
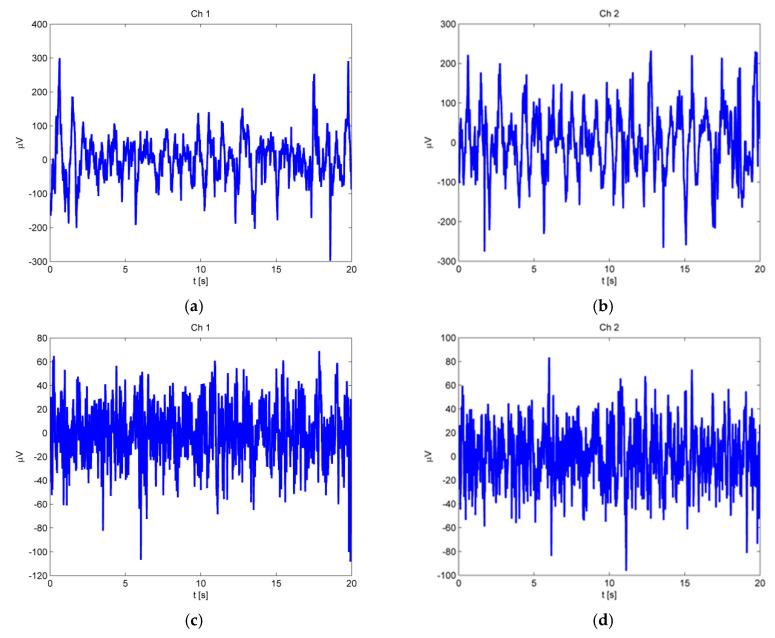
The example of bivariate signals: (**a**) focal signal from channel ch1, (**b**) focal signal from channel ch2, (**c**) non-focal signal from channel ch1, and (**d**) non-focal signal from channel ch2.

**Figure 6 entropy-23-00832-f006:**
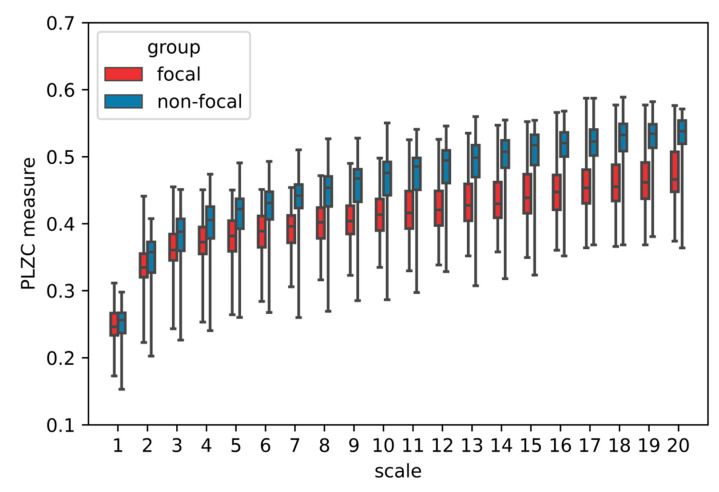
MPLZC analysis of EEG signals. The mean values of permutation Lempel–Ziv complexity measures for each group over 20 scales; m = 5.

**Figure 7 entropy-23-00832-f007:**
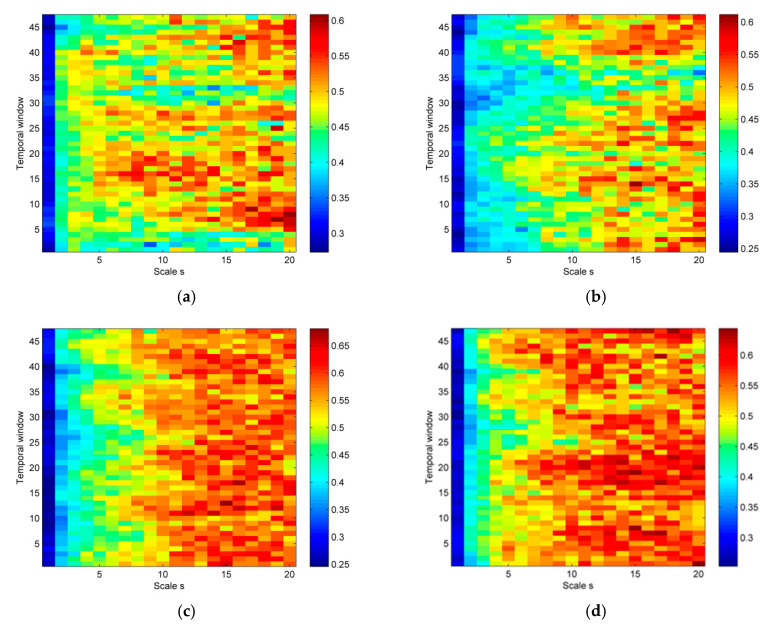
Results of the MPLZC measure on EEG signals. Relationships between: (**a**) MPLZC and focal signal from channel ch1, (**b**) MPLZC and focal signal from channel ch2, (**c**) MPLZC and non-focal signal from channel ch1, and (**d**) MPLZC and non-focal signal from channel ch2.

**Table 1 entropy-23-00832-t001:** The MPLZC results (mean ± standard deviation) of the non-parametric Mann–Whitney U test for 20 scales in studied EEG groups (* *p* < 0.05).

Measure	m	Focal Group (Mean ± std) N = 100	Non-Focal Group (Mean ± std) N = 100	*p*
PLZC_01	3	0.390 ± 0.039	0.387 ± 0.042	2.23E-01
	4	0.299 ± 0.028	0.296 ± 0.031	3.22E-01
	5	0.251 ± 0.026	0.248 ± 0.028	2.65E-01
PLZC_02	3	0.501 ± 0.046	0.503 ± 0.061	3.09E-02 *
	4	0.385 ± 0.0364	0.388 ± 0.046	1.92E-02 *
	5	0.340 ± 0.036	0.344 ± 0.044	1.54E-02 *
PLZC_03	3	0.526 ± 0.046	0.533 ± 0.064	6.96E-03 *
	4	0.404 ± 0.035	0.415 ± 0.049	3.62E-04 *
	5	0.364 ± 0.036	0.375 ± 0.048	3.11E-04 *
PLZC_04	3	0.533 ± 0.046	0.551 ± 0.063	4.80E-05 *
	4	0.413 ± 0.033	0.429 ± 0.048	2.42E-06 *
	5	0.373 ± 0.034	0.393 ± 0.049	1.22E-06 *
PLZC_05	3	0.537 ± 0.044	0.559 ± 0.059	1.69E-06 *
	4	0.418 ± 0.033	0.441 ± 0.045	5.91E-09 *
	5	0.379 ± 0.034	0.406 ± 0.048	9.22E-10 *
PLZC_06	3	0.541 ± 0.041	0.569 ± 0.057	1.90E-08 *
	4	0.424 ± 0.032	0.451 ± 0.044	1.24E-10 *
	5	0.386 ± 0.033	0.418 ± 0.046	1.45E-11 *
PLZC_07	3	0.547 ± 0.039	0.577 ± 0.055	4.26E-10 *
	4	0.429 ± 0.031	0.461 ± 0.044	2.43E-12 *
	5	0.393 ± 0.033	0.431 ± 0.047	2.52E-13 *
PLZC_08	3	0.552 ± 0.041	0.588 ± 0.052	5.36E-11 *
	4	0.434 ± 0.031	0.470 ± 0.041	2.07E-13 *
	5	0.400 ± 0.033	0.441 ± 0.046	8.31E-14 *
PLZC_09	3	0.555 ± 0.039	0.596 ± 0.052	9.69E-12 *
	4	0.441 ± 0.031	0.478 ± 0.040	7.13E-14 *
	5	0.406 ± 0.033	0.453 ± 0.045	1.79E-15 *
PLZC_10	3	0.559 ± 0.042	0.601 ± 0.050	5.07E-12 *
	4	0.447 ± 0.032	0.486 ± 0.039	1.62E-14 *
	5	0.413 ± 0.035	0.463 ± 0.045	5.78E-16 *
PLZC_11	3	0.567 ± 0.042	0.609 ± 0.049	3.83E-12 *
	4	0.452 ± 0.033	0.493 ± 0.039	2.50E-15 *
	5	0.420 ± 0.039	0.472 ± 0.042	1.73E-16 *
PLZC_12	3	0.569 ± 0.040	0.616 ± 0.048	1.05E-14 *
	4	0.456 ± 0.033	0.500 ± 0.036	1.06E-16 *
	5	0.424 ± 0.038	0.482 ± 0.040	8.91E-19 *
PLZC_13	3	0.576 ± 0.042	0.621 ± 0.049	1.90E-12 *
	4	0.464 ± 0.035	0.507 ± 0.037	5.55E-15 *
	5	0.432 ± 0.040	0.488 ± 0.043	1.04E-17 *
PLZC_14	3	0.579 ± 0.041	0.629 ± 0.050	3.84E-15 *
	4	0.470 ± 0.034	0.514 ± 0.036	2.92E-16 *
	5	0.438 ± 0.040	0.498 ± 0.041	2.23E-18 *
PLZC_15	3	0.585 ± 0.042	0.631 ± 0.047	6.95E-14 *
	4	0.473 ± 0.035	0.518 ± 0.035	9.75E-16 *
	5	0.445 ± 0.042	0.506 ± 0.041	1.05E-18 *
PLZC_16	3	0.591 ± 0.042	0.634 ± 0.047	3.78E-13 *
	4	0.479 ± 0.036	0.523 ± 0.036	1.12E-15 *
	5	0.451 ± 0.041	0.512 ± 0.039	4.04E-19 *
PLZC_17	3	0.596 ± 0.040	0.641 ± 0.045	1.31E-13 *
	4	0.485 ± 0.037	0.528 ± 0.034	3.60E-14 *
	5	0.457 ± 0.041	0.515 ± 0.040	6.80E-18 *
PLZC_18	3	0.600 ± 0.043	0.645 ± 0.046	4.02E-12 *
	4	0.490 ± 0.035	0.533 ± 0.035	7.88E-15 *
	5	0.463 ± 0.042	0.522 ± 0.040	1.36E-17 *
PLZC_19	3	0.607 ± 0.041	0.651 ± 0.043	1.47E-13 *
	4	0.493 ± 0.034	0.537 ± 0.032	3.95E-17 *
	5	0.467 ± 0.041	0.526 ± 0.038	8.48E-19 *
PLZC_20	3	0.610 ± 0.043	0.651 ± 0.043	8.29E-12 *
	4	0.497 ± 0.035	0.540 ± 0.034	2.67E-16 *
	5	0.472 ± 0.041	0.528 ± 0.038	3.74E-18 *

**Table 2 entropy-23-00832-t002:** Performance of the LS-SVM classifier.

	m	ACC (±Confidence Intervals)	SEN	SPF
MPLZC	3	0.82 (8%)	0.81	0.84
	4	0.85 (4%)	0.85	0.84
	5	0.86 (5%)	0.88	0.83

**Table 3 entropy-23-00832-t003:** The comparison of studies performed on the Bern-Barcelona database for focal and non-focal classification.

Authors (Years)	Number of Signals	Techniques Proposed	K-Fold	Accuracy
Sharma et al. (2015) [19]	50	EMD, entropy, LS-SVM	Yes	87
Sharma et al. (2015) [18]	50	DWT, entropy, Student t-test, LS-SVM	Yes	84
Sharma et al. (2017) [20]	3750	WFB, entropy, Student t-test, LS-SVM	Yes	94.25
Bhattacharyya et al. (2017) [40]	3750	TQWT, entropy, LS-SVM	Yes	84.67
Acharya et al. (2019) [24]	3750	23 features, *p*-value, LS-SVM	Yes	87.93
Gupta et al. (2019) [41]	3750	EMD, KNN entropy features, LS-SVM	Yes	83.18

## Data Availability

Not applicable.

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
