# Peer review of "Multiscale Permutation Lempel–Ziv Complexity Measure for Biomedical Signal Analysis: Interpretation and Application to Focal EEG Signals"

_entropy, 2021, doi:10.3390/e23070832_

Round 1
Reviewer 1 Report
The author of "Machine-learning-based classification of focal EEG signals using Multiscale Permutation Lempel-Ziv Complexity Measure" introduces a new metric to study brain dynamics, based on the Lempel-Ziv complexity metric. They study this metric in synthetic and biological signals, showing that it might improve the original LPC definition. However, many issues are found when reading the manuscript. I will summary the issues below, but my main conclusion is that the work does not merit publication in Entropy, mainly due to the lack of novelty of the method, and the very poor scientific design.
The first issue, which really difficulties the reading of the manuscript, is the poor English. The text needs a complete proofreading, as the grammar is extremely poor in some places, and even impossible to understand in others.
In the introduction, the authors write, between lines 58 and 90, results from a large amount of works indicating the achieved accuracies (without confidence intervals). This information is, first, too much for the introduction, and, second, unnecesary. Some of these works could be cited in the discussion, but the detailed presentation here makes no sense.
Regarding the metric itself, the novelty is low. First, the author talk about using the multiscale nature of the metric to identify different spatiotemporal patterns, but the multiscale nature only applies to the temporal domain. Moreover, the multiscale extension is, at the end of the day, a frequency-analysis. A moving average is a low-quality method to apply a low pass filter, so the proposed metrics simply consists in a filtering followed by a downsampling (to allow the identification of the sampling rate with the time constant. The same result (or even a higher quality one) would be achieved by using a low pass filter accompanied by an increase of the time constant.
When describing the synthetic signals, the author talks about white noise, colored noise, and Brownian noise. White is a color, and thus white noise fits in the second category.
Regarding the epilepsy part of the paper, it is extremely poor. The author confuses the type of crises (simple or complex) with the type of epilepsy (focal or generalized/non-focal). Moreover, the writing suggests that the same patient provides focal and non-focal recordings, what makes no sense, as the epilepsy itself (and therefore the patient) is focal or non-focal.
The description of the Bern-Barcelona database is extremely small, and no information about the signals themselves is provided at all. What kind of data is? LFP or EEG? How can there be pairs of "focal and non-focal signals"? If the data consists of 3750 pairs of recordings (whatever this means) from 5 patients, and the author only takes the first 50 pairs, how can she be sure the signals are not coming from the same subject? From the description, it is impossible to know how many issues the testing data has, but one would be that the subject and channel used for the 50 focal signals is the same, and the subject and channel used for the 50 non-focal signals is the same, so the authors are distinguishing between noise patters. However, as I indicated above, it is impossible to know, as the definition of the data is deficient in many ways. In addition, the author does not use any filtering int he data, what could explain why the low scales (non-filtered) provide no information at all while the larger scales (filtered, using a moving average filter, and then downsampled) allow for some identification.
Last, regarding the discussion, it both short and is filled with inaccuracies.
First, the sentence "Fluctuations of various waves appearing in the EEG record often treated as noise, which do not give any information about the process, were examined for statistical methods." makes no sense. No EEG waves are treated as noise, both spectral and functional connectivity analyses use all the information contained in the signal.
Moreover, the authors say that time and frequency-based methods "are widely used, but they are not optimal for electroencephalographic signals.". All the knowledge we have now of brain dynamics comes from these analyses. And the application of new metrics (like complexity), due to their convoluted definition, must be interpreted in the framework established by those older methods.
Regarding the figures:
Figure 2 does not display error bars, and the fluctuation of the metric with increasing scale suggests that the error is quite big.
Figure 3 shows a huge modulating (low frequency sine) signal. After binarization, the high frequency sine is unimportant for long time scales.
Figures 4 to 10 are extremely repetitive, and do not provide any interesting information.
The p-values shown in Table 1 are too low, at first sight, for the data shown in Figure 12. The confidence intervals are overlapping in all the cases, and that would imply a non-significant p-value.
The results shown in Table 2 area meaningless, as there are no confidence intervals in the classification scores. Even with this in mind, the scores are much lower than in most of the works shown in the second Table 2.
The second Table 2 (the index is repeated) summarize the results of other papers, supporting my suggestion that 58 to 90 in the introduction should be removed.
In summary, the title of the paper promises machine learning analysis of focal epilepsy data. The machine learning is a simple SVM classifier described in 7 lines and a table. And the focal epilepsy data is not clear at all; it is impossible to know what the authors are trying to classify. The proposed method might merit some attention, but a more careful reading indicates that there is no novelty in the approach.
Author Response
Dear Reviewer
Enclosed please find a response about manuscript by Marta Borowska, “Machine-learning-based classification of focal EEG signals using Multiscale Permutation Lempel-Ziv Complexity Measure”. I would like to take the opportunity to sincerely thank you for their comments. The paper was corrected according to the comments made by you.
Specifically:
In answer to reviewer #1 comments:
The author of "Machine-learning-based classification of focal EEG signals using Multiscale Permutation Lempel-Ziv Complexity Measure" introduces a new metric to study brain dynamics, based on the Lempel-Ziv complexity metric. They study this metric in synthetic and biological signals, showing that it might improve the original LPC definition. However, many issues are found when reading the manuscript. I will summary the issues below, but my main conclusion is that the work does not merit publication in Entropy, mainly due to the lack of novelty of the method, and the very poor scientific design.
The first issue, which really difficulties the reading of the manuscript, is the poor English. The text needs a complete proofreading, as the grammar is extremely poor in some places, and even impossible to understand in others.
The article was corrected by an English editor.
In the introduction, the authors write, between lines 58 and 90, results from a large amount of works indicating the achieved accuracies (without confidence intervals). This information is, first, too much for the introduction, and, second, unnecesary. Some of these works could be cited in the discussion, but the detailed presentation here makes no sense.
The Introduction section was rewritten:
Electroencephalography (EEG) is a non-invasive procedure to measure the bioelectric activity of the brain. The analysis of EEG signals contributes to a better understanding of brain functions and malfunctions. In particular, electroencephalographic signals from epilepsy patients are of interest to researchers.
Epilepsy is one of the most common chronic neurological diseases, characterized by recurrent seizures, manifested by disturbances in consciousness, behavior, perception, movement or sensation, which can be single or complex. According to WHO data [1], over 50 million people suffer from epilepsy, and every year on average 2.4 million patients find out that they are diagnosed with epilepsy. If an appropriate diagnosis and treatment are made, 70% of those with disease can live without seizures. The clinical picture of the seizure is the result of the discharge of improperly synchronized neurons, and it is in-fluenced by the size of the abnormally stimulated area and its location. In 2017, the cur-rent seizure classification was published [2], which includes the division into focal seizures, generalized seizures and seizures with unknown onset. In focal-onset seizures, there are seizures with the state of consciousness and without the state of consciousness, with the coexistence of motor symptoms and the development of focal seizures to bilateral tonic-clonic seizures. Among generalized seizures, there are motor seizures and non-motor seizures (absences). In the case of seizures with unknown onset, there are seizures with or without coexistence of motor symptoms, as well as unclassified seizures. The new classification noted that some tonic and flexion seizures may also have a focal onset. International League Against Epilepsy (ILAE) also presented a new classification of epilepsy and epilepsy syndromes, which takes into account the following diagnostic criteria: seizure type, epilepsy types and epilepsy syndromes [2].
The diagnosis of epilepsy patients sometimes requires EEG recording directly from the brain surface or from deeper brain structures [3]. Intracranial records can be used to locate the areas of the brain where seizures begin and to evaluate the benefits of neuro-surgical resection of these parts of the brain. It turns out that the recordings show variable dynamics not only during acute epileptic seizures, but also in the non-seizure period. There is a need to develop new methods for analyzing signals from epilepsy patients in the seizure-free period to identify them at an early stage of diagnosis to implement the appropriate treatment method as soon as possible.
Nonlinear analysis of EEG signals includes many measures that allow one to extract useful information from dynamical systems. There are many methods of detecting dy-namic changes in physiological systems, in particular some complexity indexes, such as Lempel-Ziv complexity [4], permutation Lempel-Ziv complexity [5], approximate entropy [6], sample entropy [7], fuzzy entropy [8], permutation entropy [9], multi-scale entropy [10-13], recurrence quantification analysis [14], detrended fluctuation analysis [15], fractal dimension [16], which are used as effective features of EEG signals [17-24].
Related researches have indicated that LZC [4] is powerful in analysing biomedical signals, especially in EEG analysis [25-29]. LZC based on a coarse-graining process is nonlinear measure of signal complexity and irregularity for short and nonstationary time series. Higher value of LZC means more complex structure of signal. The original Lempel-Ziv complexity algorithm consists of transformation of the signal into a binary sequence by comparing it with threshold (e.g. mean or median) and calculation of the unique subsequence in a sequence. However, the LZC measure is artifact sensitive and cannot distinguish between deterministic chaos and noise [5, 30, 31]. The binary coarse-graining process is associated with loss of signal dynamics and important system information. As an improvement of binary coarse-graining process, ordinal patterns [9] have been used to quantify dynamical information of signals. Bai et al. [5] have developed permutation Lempel-Ziv complexity measure (PLZC) to quantify dynamic changes in EEG signals. However, the existing PLZC methods have one common problem when analyzing EEG signals. EEG information is embedded in different scale domains. PLZC algorithm is a single-scale analysis and therefore it fails to account for multiple electrical activities which are inherent in the brain. The solution to this problem is to use the mul-tiscale approach to measure the complexity of EEG recordings over multiple spatiotem-poral scales of signals instead using a single scale [10]. Therefore, we proposed the mul-tiscale permutation Lempel-Ziv complexity measure (MPLZC) that combines permutation Lempel-Ziv complexity methods and a multiscale approach. The concept of multiscale Lempel-Ziv measure has already been introduced in the analysis of short, non-stationary and noisy EEG signals [32]. However, this method uses multiple thresholds for binarization, which is obtained by comparing each element of EEG signals with its smoothed versions in the window.
The purpose of the research presented in this article is to evaluate MPLZC measure in identification of focal signals.
The structure of the paper is organized as follows: Section 2 presents the description of simulated and empirical data, methods of Lempel-Ziv complexity, permutation Lempel-Ziv complexity, multiscale permutation Lempel-Ziv complexity and classifica-tion. The results are summarized in Section 3. Finally, Section 4 presents the discussion of the obtained results with corresponding analysis.
Regarding the metric itself, the novelty is low. First, the author talk about using the multiscale nature of the metric to identify different spatiotemporal patterns, but the multiscale nature only applies to the temporal domain. Moreover, the multiscale extension is, at the end of the day, a frequency-analysis. A moving average is a low-quality method to apply a low pass filter, so the proposed metrics simply consists in a filtering followed by a downsampling (to allow the identification of the sampling rate with the time constant. The same result (or even a higher quality one) would be achieved by using a low pass filter accompanied by an increase of the time constant.
The Materials and methods section was changed. The synthetic signals were analyzed in the time window. The Result section was rewritten.
When describing the synthetic signals, the author talks about white noise, colored noise, and Brownian noise. White is a color, and thus white noise fits in the second category.
Of course you are right, the description of synthetic signals was changed.
Regarding the epilepsy part of the paper, it is extremely poor. The author confuses the type of crises (simple or complex) with the type of epilepsy (focal or generalized/non-focal). Moreover, the writing suggests that the same patient provides focal and non-focal recordings, what makes no sense, as the epilepsy itself (and therefore the patient) is focal or non-focal.
The introduction section was rewritten. Epilepsy and its symptoms were described based on the latest epilepsy classification.
The description of the Bern-Barcelona database is extremely small, and no information about the signals themselves is provided at all. What kind of data is? LFP or EEG? How can there be pairs of "focal and non-focal signals"? If the data consists of 3750 pairs of recordings (whatever this means) from 5 patients, and the author only takes the first 50 pairs, how can she be sure the signals are not coming from the same subject? From the description, it is impossible to know how many issues the testing data has, but one would be that the subject and channel used for the 50 focal signals is the same, and the subject and channel used for the 50 non-focal signals is the same, so the authors are distinguishing between noise patters. However, as I indicated above, it is impossible to know, as the definition of the data is deficient in many ways. In addition, the author does not use any filtering int he data, what could explain why the low scales (non-filtered) provide no information at all while the larger scales (filtered, using a moving average filter, and then downsampled) allow for some identification.
The database description was extended with additional information.
Last, regarding the discussion, it both short and is filled with inaccuracies.
First, the sentence "Fluctuations of various waves appearing in the EEG record often treated as noise, which do not give any information about the process, were examined for statistical methods." makes no sense. No EEG waves are treated as noise, both spectral and functional connectivity analyses use all the information contained in the signal.
Moreover, the authors say that time and frequency-based methods "are widely used, but they are not optimal for electroencephalographic signals.". All the knowledge we have now of brain dynamics comes from these analyses. And the application of new metrics (like complexity), due to their convoluted definition, must be interpreted in the framework established by those older methods.
The Discussion section was rewritten.
Regarding the figures:
Figure 2 does not display error bars, and the fluctuation of the metric with increasing scale suggests that the error is quite big.
Figure 3 shows a huge modulating (low frequency sine) signal. After binarization, the high frequency sine is unimportant for long time scales.
Figures have been changed because different synthetic signals were used.
Figures 4 to 10 are extremely repetitive, and do not provide any interesting information.
Figures were removed. Other analyses were carried out, and the effect of parameters on the results was also described.
The p-values shown in Table 1 are too low, at first sight, for the data shown in Figure 12. The confidence intervals are overlapping in all the cases, and that would imply a non-significant p-value.
The figure showing the obtained results on the basis has been changed. Boxplot was made. Standard deviations are large, a description of the analysis has been added.
The results shown in Table 2 area meaningless, as there are no confidence intervals in the classification scores. Even with this in mind, the scores are much lower than in most of the works shown in the second Table 2.
The Classification section was rewritten.
The second Table 2 (the index is repeated) summarize the results of other papers, supporting my suggestion that 58 to 90 in the introduction should be removed.
Description of Table 3 was moved from Introduction to Discussion.
In summary, the title of the paper promises machine learning analysis of focal epilepsy data. The machine learning is a simple SVM classifier described in 7 lines and a table. And the focal epilepsy data is not clear at all; it is impossible to know what the authors are trying to classify. The proposed method might merit some attention, but a more careful reading indicates that there is no novelty in the approach.
Of course you are right, I will propose to change the title of the article: “Multiscale Permutation Lempel-Ziv Complexity Measure for focal EEG signals”
Reviewer 2 Report
The author combined the Lempel-Ziv complexity (LZC) measure, the permutation entropy method and the multiscale approach to propose a new complexity index called multiscale permutation Lempel-Ziv complexity measure (MPLZC), and tested the performance of the measure, and then used the least squares support vector machine (LS-SVM) to classify the focal and non-focal EEG signals obtained from five temporal lobe epilepsy patients from the Bern-Barcelona EEG database. As a result, the classification accuracy of the complexity index can reach 86%. Overall, this study first introduces the MPLZC measure to distinguish the focal and non-focal EEG signals.
Major concerns:
(1) In section 2.2 (data collection), the explanation of the data source is insufficient, and the data collection process, the number of collection channels and their distribution, etc. should be briefly explained.
(2) In this part of 2.5 (Multiscale permutation Lempel-Ziv Complexity), the variable “j” in the formula does not give a specific range. In addition, the role of τ in MPLZC is not clear. It is suggested to provide a formula containing τ, or provide a clearer explanation.
(3) In section 2.6 (classification), the explanation of the support vector machine (SVM) is insufficient, and only the concept and principle of the SVM and LS-SVM are explained. The specific use process and settings of the LS-SVM in this research should be pointed out.
(4) In section 3.2 (Neurological focal-nonfocal dataset), the preprocessing process of EEG data is not given. If no preprocessing is performed, the situation should be explained in the main text.
(5) In section 3.1 (Synthetic signals), in this part, the author has conducted a series of tests on the characteristics of MPLZC and PLZC for different signal conditions, but here the author only considers the influence of different values of the order “m” and signal length, ignores the impact of the time delay “?” and always sets the time delay “?” to 1. Here, the order “m”, the time delay “?” and the signal length should be considered together, and the selection of parameters under different signal lengths should be given.
(6) In the discussion part, there is no explanation or conjecture about the physiological and dynamic systems of the parameters' performance, that is, what is the possible cause of the change in the EEG complexity of the area when epilepsy occurs.
Minor concerns:
(1) The indentation of figures needs to medicate by the journal requirement, and the words on some figures are too small. It is highly recommended to adjust them.
(2) In Fig.2 and Fig.5~9, the contents of the figures are “the result of MPLZC analysis”, but the Y-axis labels of the figures are “PLZC”.
(3) In Fig.3 and Fig.4, the signal and noise data length of the Y-axis all use the data point “n”, but the description of time length use second “s” in the above. The two need to be unified.
(4) The positions of the legends in Fig.5-10 need to be unified, and the font size of the legends should be increased. It is best to reduce the figures’ size appropriately and put them in one line, and then attach a description of the legend below the figure.
(5) In line 349,“The result of the SVM classification is presented in Table 2” should be changed to “The result of the LS-SVM classification is presented in Table 2”

Author Response
Dear Reviewer
Enclosed please find a response about manuscript by Marta Borowska, “Machine-learning-based classification of focal EEG signals using Multiscale Permutation Lempel-Ziv Complexity Measure”. I would like to take the opportunity to sincerely thank you for their comments. The paper was corrected according to the comments made by you.
Specifically:
In answer to reviewer #2 comments:
The author combined the Lempel-Ziv complexity (LZC) measure, the permutation entropy method and the multiscale approach to propose a new complexity index called multiscale permutation Lempel-Ziv complexity measure (MPLZC), and tested the performance of the measure, and then used the least squares support vector machine (LS-SVM) to classify the focal and non-focal EEG signals obtained from five temporal lobe epilepsy patients from the Bern-Barcelona EEG database. As a result, the classification accuracy of the complexity index can reach 86%. Overall, this study first introduces the MPLZC measure to distinguish the focal and non-focal EEG signals.
Major concerns:
- In section 2.2 (data collection), the explanation of the data source is insufficient, and the data collection process, the number of collection channels and their distribution, etc. should be briefly explained.
Description of data collection was added.
- In this part of 2.5 (Multiscale permutation Lempel-Ziv Complexity), the variable “j” in the formula does not give a specific range. In addition, the role of τ in MPLZC is not clear. It is suggested to provide a formula containing τ, or provide a clearer explanation.
The formula was corrected. And a paragraph was added to Multiscale permutation Lempel-Ziv complexity subsection:
The unity delay (τ=1) is used to calculate the PLZC of the coarse-grained time series for all scales. There are several methods for determining the optimal time delay, for a single channel. Future developments should test time delay selection for different scales, but these are not yet available.
- In section 2.6 (classification), the explanation of the support vector machine (SVM) is insufficient, and only the concept and principle of the SVM and LS-SVM are explained. The specific use process and settings of the LS-SVM in this research should be pointed out.
Description of the classification was extended with additional information.
- In section 3.2 (Neurological focal-nonfocal dataset), the preprocessing process of EEG data is not given. If no preprocessing is performed, the situation should be explained in the main text.
Description of data collection was added.
- In section 3.1 (Synthetic signals), in this part, the author has conducted a series of tests on the characteristics of MPLZC and PLZC for different signal conditions, but here the author only considers the influence of different values of the order “m” and signal length, ignores the impact of the time delay “?” and always sets the time delay “?” to 1. Here, the order “m”, the time delay “?” and the signal length should be considered together, and the selection of parameters under different signal lengths should be given.
The synthetic signals were changed and an analysis was performed. The influence of the tau parameter on the results was not investigated. This is described in Multiscale permutation Lempel-Ziv complexity subsection.
- In the discussion part, there is no explanation or conjecture about the physiological and dynamic systems of the parameters' performance, that is, what is the possible cause of the change in the EEG complexity of the area when epilepsy occurs.
Discussion section was rewritten.
Minor concerns:
- The indentation of figures needs to medicate by the journal requirement, and the words on some figures are too small. It is highly recommended to adjust them.
Figures were corrected.
- In Fig.2 and Fig.5~9, the contents of the figures are “the result of MPLZC analysis”, but the Y-axis labels of the figures are “PLZC”.
Figures showed results of PLZC in 20 scales, but the method is named multiscale PLZC, hence the signature is MPLZC.
- In Fig.3 and Fig.4, the signal and noise data length of the Y-axis all use the data point “n”, but the description of time length use second “s” in the above. The two need to be unified.
All figures were presented in time domain.
- The positions of the legends in Fig.5-10 need to be unified, and the font size of the legends should be increased. It is best to reduce the figures’ size appropriately and put them in one line, and then attach a description of the legend below the figure.
New figures were done. I hope they will be visible. I saved the figures in 1000 DPI resolution, and the font was set to 8 px.
- In line 349,“The result of the SVM classification is presented in Table 2” should be changed to “The result of the LS-SVM classification is presented in Table 2”
Description was changed.
Reviewer 3 Report
the work is interesting in my opinion, but it needs some revisions on language (extensive editing), methods description (it should be reorganized and made more clear, highlighting the sequential passages used for data analysis), and statistical analysis description (the statistical analysis used results a bit unclear). The discussion is OK but it should be more focused on clinical application in some parts, in my opinion.
Author Response
Dear Reviewer
Enclosed please find a response about manuscript by Marta Borowska, “Machine-learning-based classification of focal EEG signals using Multiscale Permutation Lempel-Ziv Complexity Measure”. I would like to take the opportunity to sincerely thank you for their comments. The paper was corrected according to the comments made by you.
Specifically:
In answer to the reviewer #3 comments:
the work is interesting in my opinion, but it needs some revisions on language (extensive editing), methods description (it should be reorganized and made more clear, highlighting the sequential passages used for data analysis), and statistical analysis description (the statistical analysis used results a bit unclear). The discussion is OK but it should be more focused on clinical application in some parts, in my opinion.
The article was corrected by an English editor. All sections were rewritten and corrected. The Introduction section was rewritten, the Materials and Methods section was rewritten, and the classification subsection was completed. The synthetic signals were generated and analysed. The result section was completed about parameter selection, description of statistical analysis and classification. The Discussion section was rewritten.
Round 2
Reviewer 1 Report
The author has greatly improved the introduction section and the presentation of the results. I congratulate her because of that.
The English writing has been greatly improved, too, but it still has some weird parts, some repetitions, and some issues with the placing of the commas. The text in the abstract is written in a very simple format, with too short sentences, which give the section more a feeling of a bullet-point list than that of a redacted text.
Regarding the methodology, I still see some issues.
First, the description EEG database has been improved, but there are still some open questions.
- When the author talks about the focal/nonfocal pairs, are the two signals taken from homologous parts (for example, the contralateral area) or some completely independent areas? Do the signals come from stereotactic EEG or from surface EEG? If it is the first, I suppose all the signals comes from a reduced area of the brain.
- The author says that the signals were randomly chosen from the different participants, but then it says that only the first 50 pairs of signals were used. How can these two descriptions fit?
- The author hints in some moments the relation between the windowing-plus-averaging process and a frequency filtering, but this relation should be explicitly stated, for example around Eq. (3).
- The number of noise model evaluated is quite high, and it makes the presentation of the results confusing. As the title of the manuscript still refers to the EEG data, the importance of the simulations in the work seems unbalanced.
- The discussion ends hinting that the use of the author's approach might be used for diagnosis of focal epilepsy. This is a very bold statement from this work. First, there is no diagnosing here, but identification of a epileptogenic activity in an area (the author should consider using this word) in contrast to areas with no such activity. Second, if the data comes (I am not sure about this) from sEEG, the application in the clinic is quite limited: the only use would be, in an surgical or ICU setting, not requiring a crisis to identify the epileptogenic area. It would be far more interesting to use this approach with reconstructed source-space data from surface EEG or MEG.
Some minor issues:
- The authors still say, in the introduction, that MPLZC allows for the study of brain signal in different spatiotemporal scales (line 78). This is wrong, as the only dimension with a multiscale approach is the time.
- The description of the language used for the analyses should go in the Methods section, not the Results section.
- The acronym WGN (white Gaussian noise, I presume) is not defined.
- Some of the data in the Results section is a discussion of the results. These should better fit in the Discussion section. For example, the discussion about the effect of m in the simulations (lines 270-275).
- Both white and colored (pink) noises are listed under "white Gaussian noise" (line 161). It would be better to name this subsection just "Gaussian noise". I also fail to see why pink noise should be more regular and predictible than white noise. Maybe the authors should add a reference here, if they have one.
- The description of the rank test, if necessary (I do not think it is needed, but I do not oppose to it), should go after the introduction of the test. This is: "Mann-Whitney rank sum test (the non-parametric equivalent of t-student statistics for independent samples) we used [...]".
- The discussion section starts talking about the definition of complex systems using nonlinear differential equations. This is of no use here, as the differential equations are not used anywhere in the work (one of the advantages of information theory metrics is that they are naïve to the underlying system).
Very minor issues:
- In line 65 the authors might want to use "implies" instead of "means".
- In line 107, the word "appropriate" is not appropriate here.
- In line 342, the accuracy is presented in per-cent units, but the confidence interval is presented in per-one units. Also, the CI should also be included in Table 2.
- In line 395 the author might want to use "differentiation" instead of "identifying".

Author Response
In answer to reviewer #1 comments:
Thank you for your very critical and insightful comments. The introduced corrections increase the quality of the article.
The English writing has been greatly improved, too, but it still has some weird parts, some repetitions, and some issues with the placing of the commas. The text in the abstract is written in a very simple format, with too short sentences, which give the section more a feeling of a bullet-point list than that of a redacted text.
The article was corrected by an English editor. The Abstract section was rewritten.
Regarding the methodology, I still see some issues.
First, the description EEG database has been improved, but there are still some open questions. When the author talks about the focal/nonfocal pairs, are the two signals taken from homologous parts (for example, the contralateral area) or some completely independent areas? Do the signals come from stereotactic EEG or from surface EEG? If it is the first, I suppose all the signals comes from a reduced area of the brain. The author says that the signals were randomly chosen from the different participants, but then it says that only the first 50 pairs of signals were used. How can these two descriptions fit?
The database description was extended with additional information.
In the present work, we used Bern-Barcelona EEG database (https://www.upf.edu/web/ntsa/downloads) [3] to verify the usefulness of MPLZC measure for long-term intracranial electroencephalographic recordings (EEG) from patients suffering from pharmacoresistant focal-onset epilepsy. The clinical purpose of those recordings was to determine the brain areas to be surgically removed in each individual patient in order to achieve seizure control. The EEG data contains 3750 pairs of signals recorded from the areas of the brain where the first ictal EEG signal changes were detected (focal signals) and 3750 pairs of signals recorded from the areas of the brain that were not involved at seizure onset (non-focal signals) obtained from five temporal lobe epilepsy patients. Multichannel EEG signals were recorded with intracranial strip and depth electrodes manufactured by AD-TECH (Racine, WI,USA). All EEG signals were digitally band-pass filtered between 0.5 and 150 Hz using a fourth-order Butterworth filter. The sampling frequency was 512 Hz. The length of each signal was 10240 samples. The recordings of seizure activity and recordings made three hours after the last seizure were excluded. Each individual signal pair was selected randomly from the pool of all signals measured at focal EEG channels. This random sample was drawn without replacement and using a uniform random number generator. In the same way non-focal signals measured at non-focal EEG channels were randomly selected. Focal and non-focal signals came from independent areas of the brain. The first 50 pairs of records were used for that analysis and were made to form 100 signals in the focal and non-focal group.
The author hints in some moments the relation between the windowing-plus-averaging process and a frequency filtering, but this relation should be explicitly stated, for example around Eq. (3).
The sentence was added:
The coarse-grained process uses a procedure similar to sub-sampling.
The number of noise model evaluated is quite high, and it makes the presentation of the results confusing. As the title of the manuscript still refers to the EEG data, the importance of the simulations in the work seems unbalanced.
The title has been changed: “Multiscale Permutation Lempel-Ziv Complexity Measure for biomedical signal analysis: Interpretation and application to focal EEG signals”. The abstract has been extended to describe the analysis of synthetic signals. By introducing a new method, I wanted to show how it behaves in various situations.
Abstract: The paper analyses the complexity of electroencephalogram (EEG) signals in different temporal scales for the analysis and classification of focal and non-focal EEG signals. Futures from an original multiscale permutation Lempel-Ziv complexity measure (MPLZC) were obtained. MPLZC measure combines a multiscale structure, ordinal analysis and permutation Lempel-Ziv complexity for quantifying the dynamic changes of an electroencephalogram (EEG). We also show the dependency of MPLZC on several straight-forward signal processing concepts which appear in biomedical EEG activity via a set of synthetic signals. The main material of the study consists of EEG signals which were obtained from the Bern-Barcelona EEG database. The signals were divided into two groups: focal EEG signals (n=100) and non-focal EEG signals - (n = 100) for the statistical analysis to be performed by means of non-parametric Mann-Whitney test. The mean value of MPLZC results in the non-focal group are significantly higher than those in the focal group for scales above 1 (p<0.05). The result indicates that the non-focal EEG signals are more complex. MPLZC feature sets are used for the least squares support vector machine (LS-SVM) classifier to classify into the focal and non-focal EEG signals. Our experimental results confirmed the usefulness of the MPLZC method for distinguishing focal and non-focal EEG signals with a classification accuracy of 86%.
The discussion ends hinting that the use of the author's approach might be used for diagnosis of focal epilepsy. This is a very bold statement from this work. First, there is no diagnosing here, but identification of a epileptogenic activity in an area (the author should consider using this word) in contrast to areas with no such activity. Second, if the data comes (I am not sure about this) from sEEG, the application in the clinic is quite limited: the only use would be, in an surgical or ICU setting, not requiring a crisis to identify the epileptogenic area. It would be far more interesting to use this approach with reconstructed source-space data from surface EEG or MEG.
You are right. The end of the discussion was corrected:
The main highlights of this study are as follows:
- This is the first study to introduce a new methodology of complex systems analysis;
- The features obtained from MPLZC analysis allow for distinguishing two classes of EEG signals;
- We used only one measure so it can be useful in building a real system supporting identification of a epileptogenic activity in an area of the brain.
In the feature the proposed solution should be applied with reconstructed source-space data from surface EEG, EMG (electromyography) or EHG (electrohysterography).
Some minor issues:
The authors still say, in the introduction, that MPLZC allows for the study of brain signal in different spatiotemporal scales (line 78). This is wrong, as the only dimension with a multiscale approach is the time.
You are right, the sentence was corrected:
The solution to this problem is to use the multiscale approach to measure the complexity of EEG recordings over multiple time scales of signals instead using a single scale [10].
The description of the language used for the analyses should go in the Methods section, not the Results section.
The description of the language was moved to the Materials and Methods section.
The acronym WGN (white Gaussian noise, I presume) is not defined.
The acronym WGN was introduced in earlier line.
Some of the data in the Results section is a discussion of the results. These should better fit in the Discussion section. For example, the discussion about the effect of m in the simulations (lines 270-275).
The Parameter selection subsection was added to the Result section in order to discuss about the effect of τ and m in the analysis.
Both white and colored (pink) noises are listed under "white Gaussian noise" (line 161). It would be better to name this subsection just "Gaussian noise". I also fail to see why pink noise should be more regular and predictible than white noise. Maybe the authors should add a reference here, if they have one.
White Gaussian noise was changed to Gaussian noisy. The reference was added.
Sejdić, E., & Lipsitz, L. A. (2013). Necessity of noise in physiology and medicine. Computer methods and programs in biomedicine, 111(2), 459-470.
The description of the rank test, if necessary (I do not think it is needed, but I do not oppose to it), should go after the introduction of the test. This is: "Mann-Whitney rank sum test (the non-parametric equivalent of t-student statistics for independent samples) we used [...]".
The description of the rank test was changed.
The discussion section starts talking about the definition of complex systems using nonlinear differential equations. This is of no use here, as the differential equations are not used anywhere in the work (one of the advantages of information theory metrics is that they are naïve to the underlying system).
The sentences were changed:
The evolution of computational techniques has enabled the development of non-linear methods of analysis. These methods are derived from chaos theory and are often used in the study of complex biological systems.
Very minor issues:
- In line 65 the authors might want to use "implies" instead of "means".
- In line 107, the word "appropriate" is not appropriate here.
- In line 342, the accuracy is presented in per-cent units, but the confidence interval is presented in per-one units. Also, the CI should also be included in Table 2.
- In line 395 the author might want to use "differentiation" instead of "identifying".
Minor issues were corrected.
Reviewer 2 Report
- The order of sections needs to change correctly.
- What do the X and Y axes represent in Fig.1? Please mark them on the figure.
Author Response
In answer to reviewer #2 comments:
The article was corrected by an English editor.
1. The order of sections needs to change correctly.
The order of Materials and Methods section was changed.
2. What do the X and Y axes represent in Fig.1? Please mark them on the figure.
The axes were marked.